# Mucormycosis in a Patient with Severe COVID-19 Disease—The First Case Report in Romania and a Literature Review

**DOI:** 10.3390/medicina59030555

**Published:** 2023-03-11

**Authors:** Beatrice Mahler, Oana Andreea Parliteanu, Octavian Alexe, Corina Rîșcă Popazu, Catalina Elena Ene, Bogdan Timar, Tudor Constantinescu

**Affiliations:** 1National Institute of Pneumology Marius Nasta, 050159 Bucharest, Romania; 2Pneumology Department, University of Medicine and Pharmacy Carol Davila, 050474 Bucharest, Romania; 3Diabetes Department, Dunarea de Jos University, 800008 Galati, Romania; 4Diabetes Department, Victor Babes University of Medicine and Pharmacy, 300041 Timisoara, Romania; 5Pius Brinzeu Emergency Hospital, 300723 Timisoara, Romania

**Keywords:** COVID-19, mucormycosis, diabetes mellitus, immune system, corticosteroid therapy

## Abstract

Introduction: COVID-19 infection is often associated with a vast spectrum of opportunistic bacterial and fungal infections. Herein, we share a summary of the first case of COVID-19-associated mucormycosis (CAM) in a patient from Romania. Case presentation: A 51-year-old male non-smoker, with no known relevant medical history, who denied any previous alcohol use and was vaccinated against COVID-19 (complete scheme with Vaxzevria), was admitted to the hospital for severe COVID-19 infection. The first mucormycosis-related symptoms were reported on the eighth day of admission and were followed by the rapid deterioration of the patient’s condition and, consequently, death. The main aggravating factors, which were identified to be associated with the development of mucormycosis and with the poor outcome, were the association of severe COVID-19, new-onset COVID-19-triggered type 2 diabetes, and corticoid treatment for severe COVID-19. Conclusion: The association between severe COVID-19 and newly diagnosed type 2 diabetes, triggered by COVID-19 infection, increases the risk of severe opportunistic fungal infections and the associated mortality rates.

## 1. Introduction

This paper presents the first reported case of a patient with mucormycosis infection in a patient with a severe form of COVID-19 in Romania. Mucormycosis is an opportunistic fungal infection that usually occurs in patients with an impaired immune response and requires a set of prerequisites. An association between SARS-CoV-2 infection and mucormycosis has previously been reported, making COVID-19 a favorable condition for the onset of mucormycosis [1].

Mucormycosis has been described as a severe infection induced by mucoromycetes; it is a fungal infection that bore the name “zygomycosis” in the past. The most common “entrance gate” for these fungi is the respiratory system, although contamination can sometimes occur through an injury to the skin. After entering the body, this infection quickly spreads, affecting the sinuses before spreading to the brain and eyes. The opportunistic nature of this infection has been proven by the fact that infection requires “proper, preexisting conditions” in order to develop. The patients who are the most vulnerable to rare but severe infections are those with a poor immune defense mechanism, being linked to either diabetes mellitus, AIDS, prior organ transplantation, or immunosuppressant treatment, which includes long-term corticosteroid treatment [2,3].

During the COVID-19 pandemic, it was observed that fungal infections, including mucormycosis and invasive candidiasis, were often associated with SARS-CoV-2 infection. Between the beginning of the COVID-19 pandemic and September 2021, 2826 patients with CAM were reported in a single study, the “COSMIC study”, and another 187 patients with CAM out of 287 patients with mucormycosis were reported in another study from India [4,5].

In the “COSMIC study”, the average age of onset of CAM was 51.9 years, and the predominant gender was male (71%). Additionally, 57% of the patients required oxygen treatment due to COVID-19 infection, and 78% of the patients had diabetes. With regard to the onset of symptoms of CAM, they mainly occurred within the first 14 days from the diagnosis of COVID-19 infection (56%), while only 44% presented symptoms after 14 days [4]. In Mucov. studies 1 and 2, 65.2% of the patients who had mucormycosis also presented with CAM, totaling 187 of the 287 patients. For these patients, the most commonly associated pathology was diabetes (62.7%), and 32.6% had only a COVID-19 infection [5].

As previously stated, a “set of prerequisites” are needed for the development of this association, one of those being diabetes mellitus; 80% of the mucormycosis cases associated with COVID-19 were reported in patients with previously diagnosed diabetes mellitus. Another known risk factor is corticosteroid treatment, which was present in 76.3% of the patients with mucormycosis and COVID-19 [4]. The most frequent localization of mucormycosis was the rhino-orbital one, accounting for 56.7% out of the total 88.9% that involved the nose and sinuses [6].

In another review by Kumar et al., a more detailed account of mucormycosis affecting COVID-19 patients from April to June 2021 (third wave of COVID-19 infection) was presented. In this period, over 45,434 cases of mucormycosis infection among patients with COVID-19, leading to 4252 deaths, were reported. The highest proportion, with respect to localization, had rhino-cerebral mucormycosis (77.6%). Because mucormycosis was called “the black fungus” in the past and because of the large number of cases reported in patients with COVID-19, this association was named “the black fungus of COVID-19” [7].

No data on CAM were reported for Romania or any neighboring countries before this report.

## 2. Case Presentation

A 51-year-old man, fully vaccinated with Vaxzevria, had a positive antigen test for COVID-19 on 15 October 2021. On 20 October 2021, he was admitted to the COVID department with the following symptoms: sweating, fever, dry cough, and shortness of breath. These symptoms emerged nine days before admission and worsened in a progressive manner. The patient never smoked, denied previous use of alcohol, and, at the time of admission, had no significant medical history. At admission, his heart rate was 123 beats/min, his blood pressure was 167/115 mmHg, and his oxygen saturation was 86%, with an oxygen intake of 15 L/min delivered via a facial mask.

A physical exam revealed no significant findings, besides the consequent overall general condition of the patient, which was influenced by the COVID-19 infection.

At admission, the patient had an inflammatory syndrome, with a high white blood cell count (11,930 mm^3^; normal range = 3910–1090 mm^3^) and a high C-reactive protein level (76.61 mg/dL; normal range = 0–5 mg/dL). Additionally, he presented marked hyperglycemia (502 mg/dL; normal range = 74–106 mg/dL) despite not having a history of diabetes mellitus or dysglycemia. The total iron level at admission was 29 µg/dL (normal range = 70–180 µg/dL). No data were available on glycosylated hemoglobin, and testing for this was not available at this time in our hospital.

The chest radiography image was not consistent with the severe state that the patient was in and raised the suspicion of a pulmonary embolism; unfortunately, a CT scan was not possible at this time.

During admission, the patient received treatment according to the COVID-19 national treatment protocol: antivirals (200 mg of favipiravir meditop), corticosteroid therapy (8 mg of dexamethasone b.i.d.), anticoagulants (7500 international units of dalteparin b.i.d.), proton pump inhibitors (40 mg of omeprazole q.d.), beta-blockers (50 mg of metoprolol b.i.d.), and conversion enzyme inhibitors (10 mg of enalaprilum q.d.).

Until the eighth day of admission, the patient slightly recovered, allowing a decrease in oxygen intake from 15 to 7 L/min. After this date, his condition worsened, and he presented an SpO2 of 94% with an oxygen intake of 14 L/min delivered via a facial mask, a blood pressure of 150/90 mmHg, and a heart rate of 80 beats/min.

On the eighth day of admission, the patient complained of blurred vision, and on the ninth day, eyelid edema emerged. The suspicion of non-severe ocular inflammation was taken into consideration, and the patient received a local treatment with dexamethasone/tobramycin eye drops.

On the 10th day of admission, at 11:00 a.m., the patient’s condition rapidly deteriorated and he became comatose, not responding to verbal stimuli but responding to pain stimuli, with the base–acid balance showing severe acidosis (lactic acidosis pH < 6.633 (normal range = 7.370–7.450), lactic acid = 4.33 mmol/L (normal range = 0.50–2.20 mmol/L), glycemia = 253 mg/dL (normal range = 74–106 mg/dL)). He was transferred to the intensive care unit (ICU), where he had a Glasgow coma score of 3, systolic blood pressure of 88 mmHg, and severe metabolic acidosis (pH = 7.04 (normal range = 7.370–7.450), base deficit −26 mmol/L (normal range = calculated in relation to a normal value of base excess of −2), and lactic acid = 8.5 mmol/L (normal range = 0.50–2.20 mmol/L)). At admission to the ICU, the patient had a bruise around the left eye and at the base of the nasal pyramid, which were described during the physical exam, without a known history of trauma. The ferritin level on this date was 2722 µg/L (normal range = 20–250 µg/L).

In the ICU, the patient received insulin treatment and a vasopressor (noradrenalin) via continuous intravenous perfusion, antibiotic treatment (levofloxacin and meropenem), and antifungal treatment (400 mg of fluconazole b.i.d).

On the 13th day of admission, head and chest CT scans were performed, Figure 1 and Figure 2, from which a diagnosis of a massive ischemic stroke in the territory of the left median carotid artery and the bilateral anterior cerebral artery was made, and it converted to a hemorrhagic stroke and pneumomediastinum without surgical indications; they also revealed critical pulmonary impairment due to COVID-19 infection (over 85% of the entire pulmonary parenchyma was affected).

Due to the rapid progression of the condition in the left eye, which, by that date, was presenting eyelid necrosis and active bleeding from the eye cavity, on the 14th day of admission, Figure 3, an ophthalmology consult was requested. This consult raised the suspicion of left rhino-orbital mucormycosis, and a sample of the fluid was sent for analysis and diagnostic confirmation. For this diagnosis, molecular identification or MALDITOF tests (Matrix-assisted laser desorption/ionization-time of flight) were not used.

A neurological consult was also conducted regarding treatment for a massive stroke, and the indication was that anticoagulant treatment was necessary due to the hemorrhagic conversion of the stroke.

On this date, the 13th day of admission, the ferritin level was 5453 µg/L (normal range = 20–250 µg/L).

On the 14th day of admission, an ENT (ear, nose and throat) consult was performed via fibroscopy. This consult raised the suspicion of rhino-orbital mucormycosis, although, at this time, results were pending for bacteriological investigations that had been conducted.

Unfortunately, on the 14th day of admission, at 10:30 a.m., the patient went into cardiac arrest, and although resuscitation maneuvers were immediately started, the patient did not respond and was declared dead at 11:20 a.m.

No surgery was attempted to try to treat CAM due to the patient’s severe condition.

The results from the laboratory regarding the samples that were collected to confirm fungal infection came back positive for mucormycosis four days after the patient died, on 8 November.

## 3. Discussion

This is the first reported case of COVID-19-associated mucormycosis in Romania, a case that is consistent with the findings of Kumar et al [6,7]. The review showed that 56.28% of the 167 subjects were male, as was our patient; although it seems that this infection is more likely to affect men than women (78.9% of the 101 subjects) in the study by Kumar et al. [6], at present, we cannot be sure that this claim is still accurate.

The localization of the infection in our patient was both rhino-orbital and cerebral, causing a stroke. In early reviews, 56.7% of the localizations were rhino-orbital, while in the study conducted by Kumar et al., the most frequent site affected was rhino-cerebral (77.6%) [6,7]. Other reports have confirmed the rhino-orbital route (16%), followed by the rhino-orbital and cerebral routes (11.3%), as being the most frequent localizations of CAM [5].

The onset of CAM symptoms in our patient was similar to the onset described by Myshra et al., who reported an average onset of 17.28 ± 11.76 days ed [6]. The patient presented in this case report developed their first symptom (blurred vison) on 28 October, eight days after admission. Of the patients in Myshra et al.’s report, 59.4% had ocular symptoms, such as redness or eye pain [8]; the first symptom of the present case report was also ocular (blurred vision), followed by eyelid edema. In the same study, 93.8% of the patients presented with headaches, and 62.5% had nasal symptoms [8]; however, our patient did not experience such symptoms.

At admission, our patient had a high white blood cell count (11,930 mm^3^); in the follow-up tests, the level increased to a maximum of 41,040 mm^3^. In Myshra et al.’s study, the average white blood cell count at admission was 11,478.13 ± 369.83 mm^3^, with an even higher count in the group of patients who died due to CAM (15,775.0 ± 7650.4 mm^3^) [8]. Our results are consistent with these, with the leukocyte level being higher than in the group of patients who died, unfortunately resulting in our patient also dying.

As previously stated, mucormycosis is an opportunistic infection that needs an impaired immune system to develop. The most frequently reported comorbidity related to the onset of mucormycosis is diabetes mellitus. In a paper published by Myshra et al., 953 patients admitted for COVID-19 were studied. In this sample, 32 developed mucormycosis infections, with 87.5% of these patients having diabetes mellitus, making it the most frequent disease associated with CAM [8]. Our patient had no significant medical history prior to admission for COVID-19 infection, but he presented a high glycemic value (502 mg/dL) in his initial biological workup. Although our patient presented with metabolic acidosis, it was not ketoacidosis (CAD), but rather lactic acidosis (lactic acidosis pH < 6.633, lactic acid = 4.33 mmol/L, glycemia = 253 mg/dL). In the review mentioned above, the incidence of CAD was 14.9% [1]; this is not consistent with our patient, who did not develop CAD, but rather lactic acidosis, likely caused by a massive stroke through the mechanism of tissue hypoperfusion. In the study conducted by Myshra et al., no patients presented CAD [8].

Although our patient did not have preexisting diabetes mellitus, he most likely developed hyperglycemia due to SARS-CoV-2 infection, as shown in many studies that have proven a direct correlation between COVID-19 and the transdifferentiation of beta-cells from the pancreas [9]. The role of the Angiotensin-converting enzyme (ACE-2) receptor protein has also been proven, which binds to β-cells and induces insulin deficiency in patients with no prior history of diabetes [10]. His newly diagnosed diabetes created the prerequisites for the onset of CAM.

Another essential condition for the onset of CAM has been proven to be the use of corticosteroids. This treatment is considered to be essential in the treatment of COVID-19 infection, especially in patients who need oxygen supplementation [11]. This is the reason why it had been included in the national protocol for the treatment of severe and moderate forms of SARS-CoV-2 infection in Romania [12]. The use of corticosteroids, despite their benefits, may lead to several side effects such as induced immunosuppression, which puts the patient at risk of developing CAM [13]. Our patient received corticosteroid therapy (8 mg of dexamethasone b.i.d.) starting on the first day of admission to the hospital due to a severe form of COVID-19 infection.

In the study conducted by Ponnaiah et al., it was shown that the use of steroids and diabetes were risk factors for the onset of CAM, with diabetic patients having a 5-times higher risk of developing CAM and those receiving steroid treatment having a 3-times higher risk [14]. In our case, the patient received steroid treatment and also had a high glycemic level, including diabetic acidosis.

Low iron levels and high ferritin levels were described by Kumar et al. as being risk factors for the onset of CAM in patients with COVID-19 infection [15]. Our patient had a low level of iron and a high level of ferritin upon admission to the hospital.

It is not clear how our patient contracted CAM; an epidemiological inquiry was initiated, but no root cause was identified. In the literature, the pathways of contracting mucormycosis are mainly through the inhalation of spores, usually found in the soil, animal excrement, and decomposing organisms. The main prerequisite for developing CAM is that the host is immunocompromised [16]. Our patient had a weakened immune system due to both hyperglycemia and corticosteroid treatment, but the source of the infection was not found.

In the study conducted by Muthu et al., the authors tried to explain the epidemiology and pathophysiology of CAM in India and the rest of the world. Although the largest number of cases was reported in India (275), the authors could not find an explanation for why a higher rate of mortality was seen in the rest of the world (61.9%) versus India (36.5%). This study stated that the most common risk factors for CAM were diabetes, steroid use, and the impairment of iron metabolism, but they stated that SARS-CoV-2 (as an immunomodulatory virus) and endothelial dysfunction are still to be investigated [17].

Although a higher incidence of CAM has been reported in India, studies have indicated this condition in Europe as well. One of these studies was conducted in France by Gangneux et al., in which six patients with CAM were described, comprising 1% of the patients in the critical care unit who required mechanical ventilation due to COVID-19 [18]. Another study conducted in Germany by Seidel et al. reported 11 cases of mucormycosis, in which the average age of the patients was 57 years and most were male (61.5%). The majority of the patients required intensive care treatment for COVID-19 through mechanical ventilation [19]. Our patient was male as well, also requiring intensive care therapy and mechanical ventilation.

Besides the epidemiological inquiry conducted at the patient’s home, an investigation was conducted in the hospital, in which all patients admitted within the same timeframe as the patient reported as having mucormycosis were tested. All of the results were negative. These results are consistent with the fact that this infection is generally not transmitted from human to human [20,21].

In a study conducted by Guemas et al., a probable cause of CAM onset was described as being the construction site near the hospital, stating that the level of spores was higher due to the construction site and due to the high temperatures and humidity associated with the summer [22]. The area nearby the hospital site was investigated for construction sites, but no active construction sites were found. Additionally, our case occurred in October/November, and there were no high temperatures or high levels of humidity.

Another hospital environment risk factor for mucormycosis was described in the study conducted by Biswal et al., who discovered that the air-conditioning vents were contaminated (11.1%), while the masks that were used by the patients were also contaminated with Mucorales (1.7%) [23].

In an attempt to discover the source of CAM infection, Ghosh et al. tested the air in the patients’ rooms. They wanted to see if there were significantly higher counts of Mucorales spores in the rooms of those patients who went on to become infected. The results were positive, with the spore count being higher than in the rooms of patients who did not have CAM infections (3.55 vs. 1.5, *p* = 0.003) [24].

In our hospital, samples from the surfaces, air-conditioning installations, used face masks, and air from the rooms were tested, all of which were negative. Under these conditions, the source of CAM for the patient presented in this case report remains unknown.

## 4. Conclusions

Mucormycosis may develop in patients with severe COVID-19, with patients with comorbidities known to impair the immune system, such as diabetes mellitus, being at a higher risk. The presence of mucormycosis significantly increases the mortality rates in these patients and further complicates their overall management. Acknowledging the possible occurrence of mucormycosis in these vulnerable patients may lead to early diagnosis, followed by timely intervention, with subsequent improvements in patients’ prognoses.

## Figures and Tables

**Figure 1 medicina-59-00555-f001:**
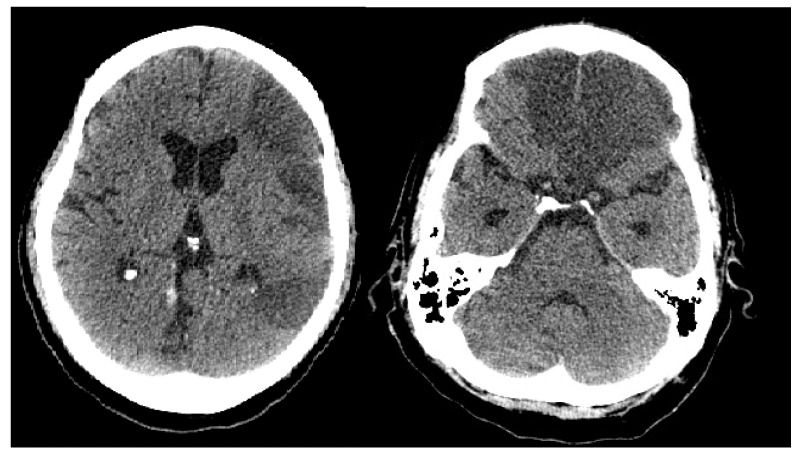
Head CT scan: Massive ischemic stroke in the territory of the left median carotid artery and the bilateral anterior cerebral artery.

**Figure 2 medicina-59-00555-f002:**
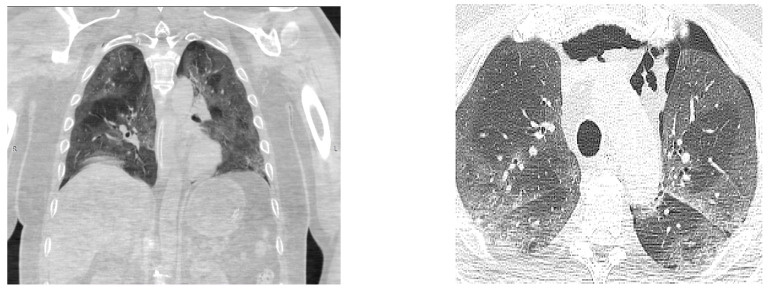
Chest CT scan: Pneumomediastinum, a critical pulmonary impairment due to COVID-19 infection.

**Figure 3 medicina-59-00555-f003:**
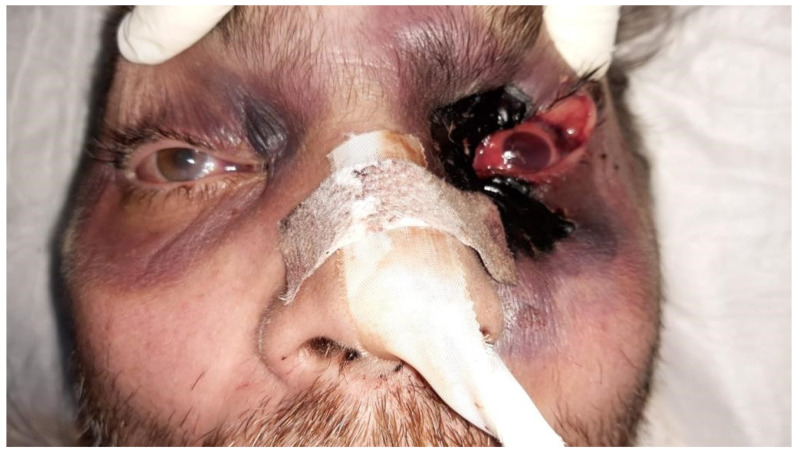
Facial presentation: Eyelid necrosis and active bleeding from the eye cavity.

## Data Availability

Not applicable.

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
