# Peer review of "Mucormycosis in a Patient with Severe COVID-19 Disease—The First Case Report in Romania and a Literature Review"

_medicina, 2023, doi:10.3390/medicina59030555_

Round 1

Reviewer 1 Report (Previous Reviewer 2)

Best regards, the authors accepted the suggestions, the manuscript seems clearer to me, it contributes to knowledge as it is an explanatory clinical case in the context of the COVID 19 pandemic, I consider it worth publishing

suggestions, the manuscript seems clearer to me, it contributes to knowledge as it is an explanatory clinical case in the context of the COVID 19 pandemic, I consider it worth publishing

Author Response

Please find the attached document.

Best regards

Reviewer 2 Report (New Reviewer)

Invasive fungal infections in patients with covid-19 infection now well-reported in the literature, with initial reports mainly emerging from India. This case report does not carry much significance except that its the first reported case from Romania.

My only comment would be why a dexamethasone dose of 8 mg twice daily was used. can you provide a reference for that? the most widely accepted dose is 6 mg once daily based on the RECOVERy trial (PMID: 32678530).

Author Response

Please find the attached document.

Best regards

This manuscript is a resubmission of an earlier submission. The following is a list of the peer review reports and author responses from that submission.

Round 1

Reviewer 1 Report

The manuscript is important since it documents the occurence of CAM in yet another geographical region. However, the manuscript needs to be edited thorughly
Do the authors have hbA1C details?
Was surgery attmepted?
Iron indices data available?
Was there any construction activity nearby?

The manuscript needs to be rewritten for grammar and the usage of terms and words:
e.g
1. line 16-17: severe-form covid 19. is it severe COVID-19?
2. Lines 19-22 abstract - apart from the grammatical part, the line does not reflect the conclusion or opinion from the current case reports and needs to be corrected
3. Line 30: "only" in patients with immune response - mucormycosis has also been reported in immune competent individuals. please rephrase
Lines 32-33 - "entrance gate"
Lines 42-44 - the lines need to be checked for grammar
Lines 45 - "pandemics"
Lines 66 - "set of prerequisites"
Line 79: Black fungus itself is a misnomer and the explanation here probabaly does not explain it and should better be avoided
In fact, the lengthy discussion does not add to it
Lines 46-79 needs to be shortened and are currently not required, further there have been more updated estimates of CAM than cited here.
notably large multicenter studies (2826 CAM patients in a single study) have reported on much larger number of patients than the 167 patients discussed here. (COSMIC study PMID 34156034 and two Mucov studies [1 and 2] PMID: 34087089).
Line 83 - ward VI may not be needed here
Lines 89-90, 96 subjective statements should preferably be modified
Lines 104-105 - needs correction
Authors should avoid using dates such as october novmeber and instead mention, day 0, day 10, and so on
Lines 139-150 - should be summarized lucidly

Discussion:
The authors have made use of two articles which have been published  much before the evaluation of CAM had taken place and several of the references are wikipedia reference
the authors should discuss the case in the context of Romania, how common is mucormycosis, other fungal infections, CAPA and diabetes
What is the status of COVID-19 associated other fungal infections in Romania and neighbouring countries?

There are atleast ten case control studies discssing the various risk factors including e.g steroids, GRP 78, diabetes, (PMID: 35939442) zinc, iron binding capacity (PMID: 34420245 PMID: 34743358 )

The Mucorales are present ubiquitously in the environment. However, the environmental presence alone is not sufficient to expalin CAM, several other factors have been discussed (PMID 34414555 )

Several countries have reported CAM and should be importantly discussed. Further the disease is not limited to India and Iran. Germany, France and several european countries (MYCOVID study france PMID: 34843666, Nation wide survey from Germany PMID: 34655486).
Construction activity near the ICU may be an important environmental clue (there are two studies to idnicate indirect relation of this PMID: 35330260, PMID: 35124141, ). The ICU environment or hospital environment may be more important than residential environment (PMID: 36118044  )

Author Response

Dear all

Please see attached form,

Thank you 

Best regards,

Oana Parliteanu

Reviewer 2 Report

The clinical case presented is of great interest because it is a complication reported in association with the COVID 19 infection reported for the first time in this geographic area, the introduction is adequate, emphasizing its objectives and the context of the findings, the antifungal therapy used is not clearly described and this is of great value in the report, the images of the patient's lesions and the imaging studies are very illustrative, it is important to recognize that we do not have mycological confirmation in this report Therefore, I would suggest that the authors include it in detail if they have it, since without this contribution the invasive fungal infection due to mucormycosis could only be classified as possible. Given that specific and standardized biomarkers have not been developed for this condition within diagnostic aids , it is of great importance to isolate the etiological agent by means of culture and to identify the genus and species , the latter due to its prognostic implication , recognizing that it is a diagnostic challenge to obtain adequate samples in severely ill patients who often present with thrombocytopenia or coagulopathy. On the other hand, it is not mentioned in the report if other diagnostic resources based on molecular identification or MALDITOF were used. It is important to clarify if this was a sporadic case and if there is no relationship with other cases in the hospital context, particularly because, although it is infrequent, cases have been reported in relation to procedures. In the discussion, it is important, in my opinion, to leave a section on the pathophysiology related to the currently proposed hypotheses that favor the appearance of this condition in the context of co-infection with COVID 19, such as the activation of the inflammatory phenotype.

Author Response

(The authors gave the same response as above.)

Round 2

Reviewer 1 Report

Poorly written.

Author Response

Dear Editor in Chief,

Dear Reviewers,

Thank you for your very important recommendations and suggestions, which, in our opinion are contributing to a significant improvement of the manuscript’s quality and scientific impact.

Based on your recommendations, we addressed all the issues raised and we aimed to correct, append or amend all these aspects. In the following, we’ll provide details of the changes added to the manuscript, in respect to your valuable comments.

Response to Reviewer 1 Comments (Round 2)

Point 1: The manuscript is important since it documents the occurence of CAM in yet another geographical region. However, the manuscript needs to be edited thorughly
Do the authors have hbA1C details?

Response 1: No, HbA1c was not determined for this patient, our hospital is a Pneumology one, and at that time it was not possible to determine HbA1c for the patients.

Point 2: Was surgery attmepted?

Response 2: No, surgery was not attempted, after the diagnosis was suspected and lab samples were drawn, there was no possibility to transfer the patient to a facility where he could have been attended surgicaly due to the critical condition of the patient. He was not in condition to be transported and unfortunatelly he died very soon after the suspiciom of CAM was reised.

Point 3: Iron indices data available?

Response 3: Yes

Total Iron was drawn on 21 october 2021 (2nd day of admission) level was 29 microgram per deciliter µg/dL  (normal range 70-180  microgram per deciliter)

Ferritin was drawn on multiple time as follows:

-on 31 october 2021 (10th day of admission) value was 2722 microgram per liter µg/L (normal range 20-250 microgram per liter µg/L)

- on 1st of november 2021 ( 11th day of admission) value was 1851 microgram per liter µg/L (normal range 20-250 microgram per liter µg/L)

-on 2nd of november (12th day of admission) value was 2156 microgram per liter µg/L (normal range 20-250 microgram per liter µg/L)

-on the 3rd of november (13th day of admission) value was 5453 microgram per liter µg/L (normal range 20-250 microgram per liter µg/L)

Point 4: Was there any construction activity nearby?

Response 4: No, at the time of this patient`s admission, there was no construction sites nearby the Hospital, in fact it was in the Pandemics perioud, in which such activities were greatly limitted.

Point 5: The manuscript needs to be rewritten for grammar and the usage of terms and words:
e.g
Line 16-17: severe-form covid 19. is it severe COVID-19?

Response 5:This was modified as per the requests stated above.

Point 6:Lines 19-22 abstract - apart from the grammatical part, the line does not reflect the conclusion or opinion from the current case reports and needs to be corrected

Response 6: It was corrected as per requirments, it is now reflecting the concusion from the current case.

Point 7:Line 30: "only" in patients with immune response - mucormycosis has also been reported in immune competent individuals. please rephrase

Response 7:It was changed from only to usually.

Point 8: Lines 32-33 - "entrance gate"

Response 8: It was modified as per the requests stated above.

Point 9: Lines 42-44 - the lines need to be checked for grammar

Response 9: It was modified to be the correct tence.

Point 10: Lines 45 - "pandemics"

Response 10: It was modified as per the requests stated above.

Point 11: Lines 66 - "set of prerequisites"

Response 11: It was modified as per the requests stated above.

Point 12: Line 79: Black fungus itself is a misnomer and the explanation here probabaly does not explain it and should better be avoided

Response 12: This sentence was rephrase as per it to explain this name.

Point 13: In fact, the lengthy discussion does not add to it
Lines 46-79 needs to be shortened and are currently not required, further there have been more updated estimates of CAM than cited here.

Response 13: It was shortened as per to still keep the inteneded logical naration of the case.

Point 14: notably large multicenter studies (2826 CAM patients in a single study) have reported on much larger number of patients than the 167 patients discussed here. (COSMIC study PMID 34156034 and two Mucov studies [1 and 2] PMID: 34087089).

Response 14: The numer of 167 cases refered here was up ti september 2021, up to the period before our case occurred, so we decided to cite this number since it was relevant for the case, it was what was known up to the poin of this case occuring in Romania.

Point 15: Line 83 - ward VI may not be needed here

Response 15: It was removed.

Point 16: Lines 89-90, 96 subjective statements should preferably be modified.

Response 16: Please explain why it is considered to be a subjective statement in this lines.

Point 17: Lines 104-105 - needs correction
Authors should avoid using dates such as october novmeber and instead mention, day 0, day 10, and so on.

Response 17: It was modified as per request.

Point 18: Lines 139-150 - should be summarized lucidly.

Response 18: This lines describe the evolutin of patient and for a better understanding of what happened and for the logical flow of showing the case, this paragraphs need to be in the current form .

Point 19: Discussion:
The authors have made use of two articles which have been published  much before the evaluation of CAM had taken place and several of the references are wikipedia reference
the authors should discuss the case in the context of Romania, how common is mucormycosis, other fungal infections, CAPA and diabetes
What is the status of COVID-19 associated other fungal infections in Romania and neighbouring countries?

Response 19: At the moment data about Romania are and were not available, hence the reason we used artcile form the literature, we also researched neighbouring countries and could not find any data available at that moment.

Point 20: There are atleast ten case control studies discssing the various risk factors including e.g steroids, GRP 78, diabetes, (PMID: 35939442) zinc, iron binding capacity (PMID: 34420245 PMID: 34743358 )

The Mucorales are present ubiquitously in the environment. However, the environmental presence alone is not sufficient to expalin CAM, several other factors have been discussed (PMID 34414555 )

Response 20: We refered at the steroid treatment and at diabets as beeing possible risck factors involved in this CAM case, in the end of our article we stated about the epidemiological enquery which was conducted at the patients house as well as at the hospital, unfortunately we were not able to isolated the exact cause of this CAM case.

Point 21: Several countries have reported CAM and should be importantly discussed. Further the disease is not limited to India and Iran. Germany, France and several european countries (MYCOVID study france PMID: 34843666, Nation wide survey from Germany PMID: 34655486).

Response 21: When writing our article, we stated about the cases that were describet at that time in europeean countries as well, as it is seen in the Introduction prt of the article “Turkey 11, Germany and UK 2, Spain, France, Italy and Austria 1 each”

Point 22: Construction activity near the ICU may be an important environmental clue (there are two studies to idnicate indirect relation of this PMID: 35330260, PMID: 35124141, ). The ICU environment or hospital environment may be more important than residential environment (PMID: 36118044  )

Response 22: In the final part of the article we stated that an inquery was conducted as wel in the hospital, trying to search for the cause of this CAM case, at that moment no constructions were taking place neither at the hospital, neither in the area surrounding the hospital. Below is the paragraph stating that.

“Besides the epidemiological inquiry conducted at patient’s home, an investigation was conducted in the hospital, by testing all patients admitted in the same timeframe with the reported patient for mucormycosis. All the results were negative. These results are consistent with the fact that this infection is generally not transmitted from human to human. [15, 16] Also tests from the surfaces and the air conditioning installations were samplet, all of those being negative. In these conditions, the source of CAM for the patient presented in this case report remains unknown.”

Reviewer 2 Report

Kind regards, the manuscript is of great interest, but I insist that it is important to clarify the antifungal therapy that was used, if it was finally used, at what dose, for how long. As well as to make clear which was the mycological diagnostic method to reach the final diagnosis conclusion. Without this, the diagnosis would only be considered as a possible diagnosis of invasive fungal infection. Since determining the genus and species of the fungus is also related to the prognosis. 

In the ICU the patient received insulin treatment and vasopressor (noradrenalin) continuous intravenous perfusion, antibiotic treatment (levofloxacin and meropenem) and antifungals treatment (levofloxacin) 

The therapy consigned here does not correspond to an antifungal, and it is not clear whether it was possible through some microbiological method to diagnose fungal infection.

Round 3

Reviewer 1 Report

No changes made in language or discussion